# Automating Active Labelling with Greedy Silhouette Search

## Abstract

Labelling data is expensive, making active learning especially valuable in low-budget settings where only a few samples can be annotated. However, existing methods often rely on delicate and complex hyper-parameter tuning, which often requires labelled validation data. We introduce Greedy Silhouette Search (GSS), a practical and robust method that leverages the Silhouette clustering metric to guide both sample selection and hyper-parameter configuration. We prove a bound on generalisation error for the 1-Nearest Neighbour (1-NN) classifier when labels are generated by GSS. Experiments demonstrate that GSS achieves competitive performance compared to baselines that require extensive tuning, making it a strong candidate for real-world, resource-constrained applications.

## 1 Introduction

Machine learning is driving significant advances across a wide range of application domains. A key factor underlying this success is the increased availability of large-scale annotated datasets in today's big data era. However, despite the abundance of raw data, obtaining high-quality labelled data remains a major bottleneck. This reliance on labelling data poses a fundamental challenge to the scalability and generalisation of traditional machine learning models.

Active learning (AL) aims to address this challenge by seeking an efficient way to selectively label the data, rather than labelling a random subset. In this process, AL progressively identifies a limited number of the most useful data points and asks an oracle to label them. Such a strategy aims at minimising the time and resources spent on annotation and helping machine learning models learn faster and more efficiently. In practice, the time and resources available for annotation can be quite limited. For example, manual annotating medical images relies on expert radiologists' or doctors' specialised knowledge Mahapatra et al. (2024). This is especially expensive and time-consuming. Quantifying the catalytic efficiency of an enzyme typically necessitates complex, highly controlled laboratory experiments conducted by trained biologists, posing challenges in terms of scalability, reproducibility, and cost Steinberg et al. (2025). Emotion analysis from textual or speech data often requires multiple human annotators to ensure label reliability, as emotional interpretation is inherently subjective and prone to inter-annotator variability Zhang et al. (2021). Numerous practical examples motivate the development of effective AL strategies in low-label-budget regimes, where only a small subset of data points can be annotated. In such a setting, the scarcity of labelled data also complicates subsequent model tuning and validation. Most of the work on AL involves hyper-parameter tuning schemes that are either sophisticated or played down Yehuda et al. (2022); Chen & Wujek (2020); Mahmood et al. (2022). Unlike that in traditional supervised learning settings, where labelled validation sets serve for this purpose, AL do not have that luxury of labelled validation sets. Therefore, delicate hyper-parameter tuning does not suit the low-budget regime of AL.

Despite being a popular evaluation metric for clustering, there is little work on directly using Silhouette in optimisation due to its prohibitive computational cost Lenssen & Schubert (2022; 2024). In this paper, we introduce Greedy Silhouette Search (GSS), which has a direct application in AL. GSS utilises the medoid Silhouette, and by developing an accelerated algorithm and adopting the greedy strategy, we make the GSS algorithm feasible in the AL setting. It serves as a sole criterion for data labelling and hyper-parameter optimisation. Combined with a 1-NN classifier, we present experiment results under a validation-set-free setting. Results on several datasets show that the proposed GSS method achieves comparable performance with competing baselines, with the advantage

of being free from hyper-parameter tuning. This shows a great potential of GSS in practice. The contributions of this paper are four-fold:

- We propose the Silhouette search algorithm for selecting samples from a dataset that optimises the macro-averaged medoid Silhouette.

- We propose the Greedy Silhouette Search (GSS) using golden-section search to simultaneously find a locally-optimal hyper-parameter, and a batch of samples efficiently under a validation-set-free setting.

- We provide and analyse the bound on the expected error of the 1-NN classifier with the proposed GSS algorithm.

- We show concretely feasible examples of directly using GSS in low-budget AL tasks with the accelerated algorithm and the greedy strategy achieving promising results.

## 2 PROBLEM SETTING AND BACKGROUND

We use AL as an example to explain the problem that GSS solves, and introduce some related work. We will sometimes use Silhouette and medoid Silhouette interchangeably for simplicity of expression.

### 2.1 PROBLEM DEFINITION

Given $\mathcal{X} \in \mathbb{R}^D$, and its true label set $\mathcal{Y} = \{1, ..., K\}$, we assume that there exists a true labelling function $f : \mathcal{X} \to \mathcal{Y}$. In the AL setting, a small labelled dataset $\mathcal{L} = \{\tilde{\mathbf{x}}_l, \tilde{y}_l\}_{l=1}^{L}$ is made available initially, where $L$ is the number of labelled samples that is usually very small or can be zero. The aim of AL is to select samples from a considerably larger pool of unlabelled instances, $\mathcal{U} = \{\mathbf{x}_u\}_{u=1}^{U}$, for labelling by an oracle based on a predefined query strategy in an iterative manner, until a fixed budget of labels, $B$, are obtained. With a low budget, the selected samples may not be able to appropriately represent the underlying data distribution, so the predictor trained from them is likely unreliable. Query strategies that do not rely on the predictor are often more robust Hacohen et al. (2022).

#### 2.1.1 OBJECTIVE

The goal of AL is to find a query strategy $q$ to identify an instance $\mathbf{x}_u$ in the unlabelled pool $\mathcal{U}$ for an oracle to annotate, in order to optimise the predictor $\hat{f} : \mathcal{X} \to \hat{\mathcal{Y}}$ based on the set $\mathcal{L} \cup \{\mathbf{x}_u\}$, where $\mathcal{L}$ is a given set of labelled points that can be empty initially, and $\hat{\mathcal{Y}}$ denotes the set of predicted labels:

$$\mathbf{x}_u = q(\mathcal{U}, \mathcal{L}; \gamma) \tag{1}$$

In the low-budget regime of AL that is the focus of this work, we posit that we are unlikely to have a labelled validation set for reliably tuning the query strategy hyper-parameters $\gamma$, or subsequent model hyper-parameters. Therefore, our objective is to find an efficient query strategy $q$ for AL in a low-budget regime that uses a self-contained criterion for both hyper-parameter search and instance selection — saving us from requiring a potentially costly validation set.

### 2.2 RELATED WORK

Fig. 1 shows the overlap among two popular problems: clustering and AL. Clustering intersects with AL when it is done incrementally or in a greedy way regarding adding new cluster centres. Some Herding-based algorithms Bae et al. (2024; 2025); Yehuda et al. (2022) belong to this intersection.

#### 2.2.1 SILHOUETTE

Silhouette Rousseeuw (1987) is a method of interpreting and validating the consistency within clusters of data. It is a popular performance metric for evaluating clustering methods or a tool for choosing the optimal number of clusters. Unfortunately, directly finding a solution that maximises the Silhouette is computationally prohibitive, even with the simplified medoid Silhouette Van der

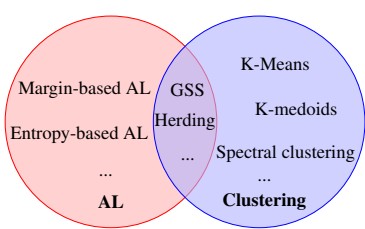
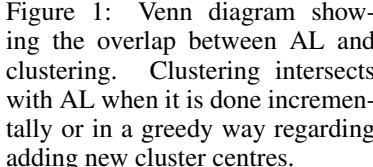

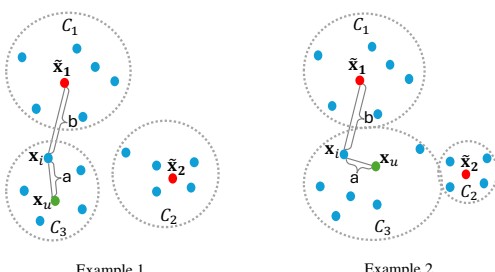

Figure 1: Venn diagram showing the overlap between AL and clustering. Clustering intersects with AL when it is done incrementally or in a greedy way regarding adding new cluster centres.

Figure 2: Two examples of calculating Equation (4) for sample $\mathbf{x}_i$ with different candidates $\mathbf{x}_u$. The red dots represent the labelled points, and the blue dots are the unlabelled points. The green dots are the currently selected $\mathbf{x}_u$, which gathers a group of unlabelled data $\mathcal{C}_3$. Choosing different $\mathbf{x}_u$ results in different configurations and scores for the sets $\mathcal{C}_l$.

Laan et al. (2003); Lenssen & Schubert (2024; 2022). As a combinatorial problem with no closed-form solution, it also needs to calculate and store pairwise distances. The complexity remains the bottleneck of its application.

### 2.2.2 INTERACTIVE ACTIVE LEARNING

We can categorise AL methods into interactive methods and model-agnostic methods. Interactive approaches Haimovich et al. (2024); Bressan et al. (2024); Wu et al. (2022) rely on task-specific model predictions/uncertainties to guide sample selection. Most of the interactive approaches rely on feedback from downstream models other than clustering methods. For example, margin-based AL selects the sample for which the classifier is least confident in its prediction by measuring the margin between the top two predicted class probabilities Bressan et al. (2024); Wang & Singh (2016). Entropy-based AL measures uncertainty using the entropy of the predicted probability distribution Wu et al. (2022); Siddiqui et al. (2020). When labelled data are scarce, model feedback often fails to provide more useful information than noise. Interactive methods are primarily effective in high-budget settings Hacohen et al. (2022) combined with complex parametric models.

### 2.2.3 MODEL-AGNOSTIC ACTIVE LEARNING

Model-agnostic methods often outperform interactive methods in a low-budget regime Bae et al. (2024; 2025), and are sometimes combined with non-parametric clustering methods as downstream task-oriented models. It does not use information from downstream model outputs. Model-agnostic methods aim to select examples from the unlabelled pool that are most representative of the underlying data distribution, under the assumption that strong performance on these "typical" examples will generalise effectively to the broader unseen dataset. Some methods formulate the problem as maximising their defined "coverage". For example, ProbCover Yehuda et al. (2022) seeks a labelled set that maximises the proposed "coverage" defined by balls centred at these labelled points. The method is however highly sensitive to the heuristically chosen ball radius Bae et al. (2024). MaxHerding Bae et al. (2024) proposes a smoother Gaussian similarity-based coverage, while introducing a less sensitive hyper-parameter of the Gaussian kernel radius. Our algorithm can be considered to be in this class; however, we do not rely on a labelled validation set. Combined with a 1-NN classifier, we achieve a validation-set-free setting. One important process in AL methods that often gets downplayed is hyper-parameter tuning. There is a lack of a consistent way to tune hyper-parameters, and low-budget setting amplifies this caveat.

## 3 PROPOSED ALGORITHMS

Model-agnostic AL often relies on a good measure of data density. Inspired by a popular performance metric for clustering, Silhouette Vendramin et al. (2010), we propose the GSS. We adopt a greedy algorithm to maximise the aggregated medoid Silhouette. Each point will have an associated Silhouette value; the final metric is given by appropriately aggregating them. A traditional way to do so is through a micro-average; that is simply calculating the mean of Silhouette values over all data points. We however use a macro-average strategy for aggregation as it has been shown empirically to be more robust against varying data distribution Pavlopoulos et al. (2024). The Silhouette values are

firstly averaged within each cluster, and then the results are averaged over clusters. In the AL scenario, we progressively add an instance to the labelled set for generating the largest macro-averaged medoid Silhouette, one by one, until we reach the budget limit $B$.

Each unlabelled point $\mathbf{x}_u$ is associated with a score $S_u$, which is the macro-averaged medoid Silhouette score. Our main algorithm is to label a point maximising the macro-averaged medoid Silhouette:

$$S_u = \frac{1}{L+1} \sum_{l=1}^{L+1} S(\mathcal{C}_l), \tag{2}$$

where $\mathcal{C}_l$ is a set of unlabelled data points $\{\mathbf{x}_i\}_{i=1}^{|\mathcal{C}_l|}$, in which each sample $\mathbf{x}_i$ satisfies: $\arg\min_{\tilde{\mathbf{x}} \in \mathcal{L}} d_\gamma(\mathbf{x}_i, \tilde{\mathbf{x}}) = \tilde{\mathbf{x}}_l$. $|\mathcal{C}_l|$ is the number of samples in the set $\mathcal{C}_l$. $\tilde{\mathbf{x}}_l$ is the $l$-th sample in the labelled data set $\mathcal{L}$. $d_\gamma(\cdot, \cdot)$ is a distance function for two samples parametrised by $\gamma$. The medoid Silhouette $S(\mathcal{C}_l)$ for each $\mathcal{C}_l$ is defined as the average medoid Silhouette for all instances in the cluster $\mathcal{C}_l$:

$$S(\mathcal{C}_l) = \frac{1}{|\mathcal{C}_l|} \sum_{\mathbf{x}_i \in \mathcal{C}_l} s(\mathbf{x}_i). \tag{3}$$

The medoid Silhouette for an instance $\mathbf{x}_i$ is formulated as:

$$s(\mathbf{x}_i) = 1 - \frac{a(\mathbf{x}_i)}{b(\mathbf{x}_i)}, \quad \text{s.t. } a(\mathbf{x}_i) = \min_{\tilde{\mathbf{x}} \in \mathcal{L}} d_\gamma(\mathbf{x}_i, \tilde{\mathbf{x}}), \ b(\mathbf{x}_i) = \min_{\tilde{\mathbf{x}} \in \mathcal{L}}{}_2\, d_\gamma(\mathbf{x}_i, \tilde{\mathbf{x}}), \tag{4}$$

where $\min_2$ denotes the operator that returns the second-smallest value. In other words, the medoid Silhouette for an instance is defined by its distance to the closest and second-closest labelled samples. From the above definition, we know $0 \le s(\mathbf{x}_i) \le 1$, and a larger Silhouette encourages larger gaps between different $\mathcal{C}_l$ and points gather tighter within each $\mathcal{C}_l$. Figure 2 shows two examples for the calculation of Equation (4) for a sample $\mathbf{x}_i$. A naive algorithm for searching a point maximising the macro-averaged medoid Silhouette is detailed in Algorithm 1.

### 3.1 COMPLEXITY ANALYSIS AND THE ACCELERATED ALGORITHM

In practical applications such as AL, Algorithm 1 is performed iteratively until a prefixed budget is reached. Since the complexity $\mathcal{O}(DU^2)$ on calculation of pairwise distance is inevitable, we do not include this term in the analysis here. Despite using a simplified version of Silhouette, Algorithm 1

---

**Algorithm 1** Silhouette Search

**Input**: $\gamma$, $\mathcal{L} = \{\tilde{\mathbf{x}}_l\}_{l=1}^L$, $\mathcal{U} = \{\mathbf{x}_u\}_{u=1}^U$,
**Output**: $\mathbf{x}^* \in \mathcal{U}, S_\gamma^*$.
1: **for** $\mathbf{x}_u \in \mathcal{U}$ **do**
2:     **for** $\mathbf{x}_i \in \mathcal{U} \backslash \mathbf{x}_u$ **do**
3:         $a(\mathbf{x}_i) = \min\limits_{\tilde{\mathbf{x}} \in \mathcal{L} \cup \mathbf{x}_u} d_\gamma(\mathbf{x}_i, \tilde{\mathbf{x}})$,
4:         $b(\mathbf{x}_i) = \min\limits_{\tilde{\mathbf{x}} \in \mathcal{L} \cup \mathbf{x}_u}{}_2\, d_\gamma(\mathbf{x}_i, \tilde{\mathbf{x}})$.
5:     **end for**
6:     $S(\mathcal{C}_l) = \frac{1}{|\mathcal{C}_l|} \sum\limits_{\mathbf{x}_i \in \mathcal{C}_l} 1 - \frac{a(\mathbf{x}_i)}{b(\mathbf{x}_i)}$
7:     $\mathcal{C}_l = \{\mathbf{x}_i | \arg\min\limits_{\tilde{\mathbf{x}} \in \mathcal{L} \cup \mathbf{x}_u} d_\gamma(\mathbf{x}_i, \tilde{\mathbf{x}}) = \tilde{\mathbf{x}}_l\}$
8:     $S_u = \frac{1}{L+1} \sum\limits_{l=1}^{L+1} S(\mathcal{C}_l)$
9: **end for**
10: $\mathbf{x}^* = \arg\max\limits_{\mathbf{x}_u \in \mathcal{U}} S_u$.
11: $S_\gamma^* = \max\limits_{\mathbf{x}_u \in \mathcal{U}} S_u$.

**Algorithm 2** Silhouette Search (Accelerated)

**Input**: $\gamma$, $\mathcal{L} = \{\tilde{\mathbf{x}}_l\}_{l=1}^L$, $\mathcal{U} = \{\mathbf{x}_u\}_{u=1}^U$, Distance Matrix $\mathbf{D} \in \mathbb{R}^{U \times U}$ for the labelled points in $\mathcal{L}$,
**Output**: $\mathbf{x}^* \in \mathcal{U}, S_\gamma^*$.

1: $\forall \mathbf{x}_u \in \mathcal{U}$, calculate:
$$\mathbf{d}^{(1)} = [d_1^{(1)}, ..., d_U^{(1)}]^\top, d_i^{(1)} = \min_{\tilde{\mathbf{x}} \in \mathcal{L}} d_\gamma(\mathbf{x}_i, \tilde{\mathbf{x}}),$$
$$\mathbf{d}^{(2)} = [d_1^{(2)}, ..., d_U^{(2)}]^\top, d_i^{(2)} = \min_{\tilde{\mathbf{x}} \in \mathcal{L}}{}_2\, d_\gamma(\mathbf{x}_i, \tilde{\mathbf{x}}).$$
2: $\mathbf{D} = \min(\mathbf{D}, \mathbf{d}^{(2)}\mathbf{1}^\top)$     ▷ element-wise min
3: $\mathbf{S} = 1 - \min(\frac{\mathbf{D}}{\mathbf{d}^{(1)}\mathbf{1}^\top}, \frac{\mathbf{d}^{(1)}\mathbf{1}^\top}{\mathbf{D}})$   ▷ element-wise division
4: $\forall u$, calculate $S_u$ using $\mathbf{S}$,
5: $\mathbf{x}^* = \arg\max\limits_{\mathbf{x}_u \in \mathcal{U}} S_u$.
6: $S_\gamma^* = \max\limits_{\mathbf{x}_u \in \mathcal{U}} S_u$.

---

takes $\mathcal{O}(LU^2)$ time with $U$ unlabelled and $L$ labelled points, to select just one sample. Noting that we only need to update the medoid Silhouette for a point when the new labelled point is within the top-2 closest points of it, we propose an accelerated version of Algorithm 1 as described in Algorithm 2. This helps reduce the complexity of Algorithm 1 to $\mathcal{O}(U^2)$. Line 2 of Algorithm 2 helps identify points that are within the top-2 closest points of each row, and then line 3 calculates medoid Silhouette for all samples in the unlabelled set using the fact that medoid Silhouette values are upper bounded by 1. $i$-th column of $\mathbf{S}$ gives the medoid Silhouette values for all unlabelled samples, when $\mathbf{x}_i$ in the unlabelled set is selected as a new instance to label. Macro-average aggregation is also applied in line 4, similar to Algorithm 1 using Eq. equation 2 and Eq. equation 3. We report the comparison of running time in the Appendix.

## 3.2 GOLDEN-SECTION SEARCH STRATEGY FOR OPTIMISATION WITH HYPER-PARAMETER

---

**Algorithm 3** Golden-Section Search

---

**Input:** Interval $[a, b]$, tolerance $\varepsilon > 0$, the golden ratio's reciprocal $r = 0.618$.
**Output:** Optimal $\mathbf{x}^*$ with $\gamma \in [a, b]$.
**Initialise:** $c = b - r(b - a)$, $d = a + r(b - a)$
**Evaluate:** $S_c^*$ and $S_d^*$ with Algo. 2

1: **while** $|b - a| > \varepsilon$ **do**
2:     **if** $S_c^* > S_d^*$ **then**
3:         Set $b \leftarrow d$
4:         Set $d \leftarrow c$, $S_d^* \leftarrow S_c^*$
5:         Set $c \leftarrow b - r(b - a)$, evaluate $S_c^*$ with Algo. 2
6:     **else**
7:         Set $a \leftarrow c$
8:         Set $c \leftarrow d$, $S_c^* \leftarrow S_d^*$
9:         Set $d \leftarrow a + r(b - a)$, evaluate $S_d^*$ with Algo. 2
10:     **end if**
11: **end while**
12: let $\gamma = \frac{a+b}{2}$, obtain $\mathbf{x}^*$ and $S_\gamma^*$ with Algo. 2.

---

It is common that the distance function will involve some hyper-parameters, like $\gamma$. In the low-budget regime AL, obtaining a labelled validation set is infeasible. As a result, conventional hyper-parameter tuning methods, which typically rely on such validation data, are ill-suited for AL under this constraint. The medoid Silhouette naturally acts as a selection criterion for selecting the $\gamma$, eliminating the need for a labelled validation set.

Because of the discreteness of medoid Silhouette, we leverage the golden-section search algorithm to search for the locally optimal hyper-parameter $\gamma$ together with the best instance to label. The hyper-parameter is optimised by gradually narrowing down the query interval to where the locally optimal value sits in. The algorithm is detailed in Algorithm 3.

## 3.3 GREEDY SILHOUETTE SEARCH (GSS) ALGORITHM

Each round of AL could involve the selection of a set of $T$ instances instead of one. In this scenario, Algorithm 2 could be run $T$ times to obtain a sum of Silhouette, and Algorithm 3 is implemented in a way that minimises this sum selecting $T$ instances instead of one. The instances are selected one-after-one in a greedy manner, hence the name "Greedy" Silhouette Search. Such a greedy strategy makes using Silhouette as a guidance for active labelling feasible. Being a combinatorial problem, considering multiple instances to optimise Silhouette jointly and iteratively with hyper-parameters is computationally infeasible. In a multiple-round AL setting, the GSS is able to choose different hyper-parameters in each round, reflecting the change of number on labelled instances better compared to methods with static hyper-parameters.

## 3.4 THEORETICAL ANALYSIS

We analyse the bound on the expected error of the 1-NN classifier with the proposed GSS algorithm. 1-NN operates in a fully non-parametric fashion, depending only on distances to the labelled set, with no supplementary inductive bias introduced.

**Theorem 1.** *Let $d$ be a distance function and $\min_2$ denotes the operator that returns the second-smallest value. $\forall \theta \in [0, 1]$, the expected error of the 1-NN classifier $\hat{f}$ is bounded as:*

$$\mathbb{E}[\mathbb{1}_{f(\mathbf{x}) \neq \hat{f}(\mathbf{x})}] \leq (1 - \Pr_{\mathbf{x}}(\frac{\min_{\tilde{\mathbf{x}} \in \mathcal{L}} d(\mathbf{x}, \tilde{\mathbf{x}})}{\min_2 \limits_{\tilde{\mathbf{x}} \in \mathcal{L}} d(\mathbf{x}, \tilde{\mathbf{x}})} \leq \theta)) + \Pr_{\mathbf{x}}(\frac{\min_{\tilde{\mathbf{x}}: f(\mathbf{x}) \neq f(\tilde{\mathbf{x}})} d(\mathbf{x}, \tilde{\mathbf{x}})}{\min_2 \limits_{\tilde{\mathbf{x}} \in \mathcal{L}} d(\mathbf{x}, \tilde{\mathbf{x}})} \leq \theta). \tag{5}$$

*Proof.*

$$\mathbb{E}[1_{f(\mathbf{x}) \neq \hat{f}(\mathbf{x})}] = \Pr_{f(\mathbf{x}) \neq \hat{f}(\mathbf{x})} \left( \frac{\min\limits_{\tilde{\mathbf{x}} \in \mathcal{L}} d(\mathbf{x}, \tilde{\mathbf{x}})}{\min\limits_{\tilde{\mathbf{x}} \in \mathcal{L}} 2 \, d(\mathbf{x}, \tilde{\mathbf{x}})} > \theta \right) + \Pr_{f(\mathbf{x}) \neq \hat{f}(\mathbf{x})} \left( \frac{\min\limits_{\tilde{\mathbf{x}} \in \mathcal{L}} d(\mathbf{x}, \tilde{\mathbf{x}})}{\min\limits_{\tilde{\mathbf{x}} \in \mathcal{L}} 2 \, d(\mathbf{x}, \tilde{\mathbf{x}})} \leq \theta \right)$$

$$\leq \Pr_{\mathbf{x}} \left( \frac{\min\limits_{\tilde{\mathbf{x}} \in \mathcal{L}} d(\mathbf{x}, \tilde{\mathbf{x}})}{\min\limits_{\tilde{\mathbf{x}} \in \mathcal{L}} 2 \, d(\mathbf{x}, \tilde{\mathbf{x}})} > \theta \right) + \Pr_{f(\mathbf{x}) \neq \hat{f}(\mathbf{x})} \left( \frac{\min\limits_{\tilde{\mathbf{x}} \in \mathcal{L}} d(\mathbf{x}, \tilde{\mathbf{x}})}{\min\limits_{\tilde{\mathbf{x}} \in \mathcal{L}} 2 \, d(\mathbf{x}, \tilde{\mathbf{x}})} \leq \theta \right).$$

For any given $\mathbf{x}$ , there are two cases:

1. $f(\mathbf{x}) = \hat{f}(\mathbf{x})$. In this case, $\Pr_{f(\mathbf{x}) \neq \hat{f}(\mathbf{x})} \left( \frac{\min\limits_{\tilde{\mathbf{x}} \in \mathcal{L}} d(\mathbf{x}, \tilde{\mathbf{x}})}{\min\limits_{\tilde{\mathbf{x}} \in \mathcal{L}} 2 \, d(\mathbf{x}, \tilde{\mathbf{x}})} \leq \theta \right) = 0$, and $\Pr_{\mathbf{x}} \left( \frac{\min\limits_{\tilde{\mathbf{x}}: f(\mathbf{x}) \neq f(\tilde{\mathbf{x}})} d(\mathbf{x}, \tilde{\mathbf{x}})}{\min\limits_{\tilde{\mathbf{x}} \in \mathcal{L}} 2 \, d(\mathbf{x}, \tilde{\mathbf{x}})} \leq \theta \right) \geq 0$. We have:

$$\Pr_{f(\mathbf{x}) \neq \hat{f}(\mathbf{x})} \left( \frac{\min\limits_{\tilde{\mathbf{x}} \in \mathcal{L}} d(\mathbf{x}, \tilde{\mathbf{x}})}{\min\limits_{\tilde{\mathbf{x}} \in \mathcal{L}} 2 \, d(\mathbf{x}, \tilde{\mathbf{x}})} \leq \theta \right) \leq \Pr_{\mathbf{x}} \left( \frac{\min\limits_{\tilde{\mathbf{x}}: f(\mathbf{x}) \neq f(\tilde{\mathbf{x}})} d(\mathbf{x}, \tilde{\mathbf{x}})}{\min\limits_{\tilde{\mathbf{x}} \in \mathcal{L}} 2 \, d(\mathbf{x}, \tilde{\mathbf{x}})} \leq \theta \right).$$

2. $f(\mathbf{x}) \neq \hat{f}(\mathbf{x})$. Let $\mathbf{n} \in \mathcal{L}$ denotes the nearest neighbour to sample $\mathbf{x}$, then we have the predicted labelled $\hat{f}(\mathbf{x}) = \hat{f}(\mathbf{n})$. Because $\mathbf{n}$ is in the labelled set, thus we have $\hat{f}(\mathbf{n}) = f(\mathbf{n})$. Since $f(\mathbf{x}) \neq \hat{f}(\mathbf{x})$, which implies that $\hat{f}(\mathbf{x}) = \hat{f}(\mathbf{n}) = f(\mathbf{n}) \neq f(\mathbf{x})$. Thus,

$$\Pr_{f(\mathbf{x}) \neq \hat{f}(\mathbf{x})} \left( \frac{\min\limits_{\tilde{\mathbf{x}} \in \mathcal{L}} d(\mathbf{x}, \tilde{\mathbf{x}})}{\min\limits_{\tilde{\mathbf{x}} \in \mathcal{L}} 2 \, d(\mathbf{x}, \tilde{\mathbf{x}})} \leq \theta \right) = \Pr_{\mathbf{x}} \left( \frac{\min\limits_{\tilde{\mathbf{x}} \in \mathcal{L}, f(\mathbf{x}) \neq f(\tilde{\mathbf{x}})} d(\mathbf{x}, \tilde{\mathbf{x}})}{\min\limits_{\tilde{\mathbf{x}} \in \mathcal{L}} 2 \, d(\mathbf{x}, \tilde{\mathbf{x}})} \leq \theta \right) \leq \Pr_{\mathbf{x}} \left( \frac{\min\limits_{\tilde{\mathbf{x}}: f(\mathbf{x}) \neq f(\tilde{\mathbf{x}})} d(\mathbf{x}, \tilde{\mathbf{x}})}{\min\limits_{\tilde{\mathbf{x}} \in \mathcal{L}} 2 \, d(\mathbf{x}, \tilde{\mathbf{x}})} \leq \theta \right).$$

Combine the two cases, and we conclude the proof:

$$\mathbb{E}[1_{f(\mathbf{x}) \neq \hat{f}(\mathbf{x})}] \leq \Pr_{\mathbf{x}} \left( \frac{\min\limits_{\tilde{\mathbf{x}} \in \mathcal{L}} d(\mathbf{x}, \tilde{\mathbf{x}})}{\min\limits_{\tilde{\mathbf{x}} \in \mathcal{L}} 2 \, d(\mathbf{x}, \tilde{\mathbf{x}})} > \theta \right) + \Pr_{\mathbf{x}} \left( \frac{\min\limits_{\tilde{\mathbf{x}}: f(\mathbf{x}) \neq f(\tilde{\mathbf{x}})} d(\mathbf{x}, \tilde{\mathbf{x}})}{\min\limits_{\tilde{\mathbf{x}} \in \mathcal{L}} 2 \, d(\mathbf{x}, \tilde{\mathbf{x}})} \leq \theta \right)$$

$$= \left( 1 - \Pr_{\mathbf{x}} \left( \frac{\min\limits_{\tilde{\mathbf{x}} \in \mathcal{L}} d(\mathbf{x}, \tilde{\mathbf{x}})}{\min\limits_{\tilde{\mathbf{x}} \in \mathcal{L}} 2 \, d(\mathbf{x}, \tilde{\mathbf{x}})} \leq \theta \right) \right) + \Pr_{\mathbf{x}} \left( \frac{\min\limits_{\tilde{\mathbf{x}}: f(\mathbf{x}) \neq f(\tilde{\mathbf{x}})} d(\mathbf{x}, \tilde{\mathbf{x}})}{\min\limits_{\tilde{\mathbf{x}} \in \mathcal{L}} 2 \, d(\mathbf{x}, \tilde{\mathbf{x}})} \leq \theta \right).$$

$\square$

**Discussion** Theorem 1 shows that minimising the medoid Silhouette value contributes to a better classification performance with 1-NN. The first term in the RHS of the inequality is related to the medoid Silhouette to be optimised, and the second term is non-increasing with adding labelled samples. A term similar to $\min\limits_{\tilde{\mathbf{x}}: f(\mathbf{x}) \neq f(\tilde{\mathbf{x}})} d(\mathbf{x}, \tilde{\mathbf{x}})$ also appears in theoretical findings of some related work Yehuda et al. (2022); Bae et al. (2024), but this term prevents the error bound therein being asymptotic to zero with the number of labelled points goes to infinity. Our bound shows such an asymptotic property and is thus more practical. More details are in the Appendix.

## 4 EXPERIMENTS

We apply the GSS on AL learning tasks to verify its effectiveness and compare it with a few state-of-the-art model-agnostic AL methods. We follow the previous work Bae et al. (2024) to measure classification performance using a 1-NN classifier. This aligns with the low-budget active learning regime, where the model validation is difficult. Ablation study shows the usefulness of each design element.

**Datasets** We evaluate our algorithms on low-budget AL across four diverse classification datasets: *OptDigits* Alpaydin & Kaynak (1998), *CIFAR10* Krizhevsky et al. (2009), *TrpB* Johnston et al. (2024), and *Phishing Websites* Mohammad & McCluskey (2012): (1) *OptDigits*[1]: a collection of

---

[1]The OptDigits dataset is available at the UCI Machine Learning Repository: https://archive.ics.uci.edu/.

hand-written digit images. It includes 5620 images with 64 features extracted from 32x32 bitmaps and 10 classes. (2) *CIFAR10*: a widely-used dataset for image classification in 10 classes. We follow previous work Yehuda et al. (2022) and use their published processed features. (3) *TrpB*[2]: a protein sequence dataset with 2 classes, however being quite imbalanced (the ratio of the number of positive and negative classes is around 1:20). We randomly filter negative samples for a balanced dataset containing 20280 samples in total. We then embed the features using one-hot encoding. (4) *Phishing Websites*[3]: a tabular dataset for phishing websites recognition consisting of two classes.

**Implementation Details**  We set the budget size at each round to 10 for two relatively small datasets and 10/100 for the others. We adopt a 1-NN classifier based on the labelled set $\mathcal{L}_t$ at $t$-th iteration, where the predicted label of a sample $\mathbf{x}$ is the label of its nearest point in $\mathcal{L}_t$. This aligns with the low-budget active learning regime where the number of available annotated samples is very small, and so model validation becomes difficult. We notice that some work uses deep neural networks such as ResNet-18 as a classifier, but this usually involves a lot of (at least 10% of training data) labelled data for validation. Unfortunately, it is often not practical under the low-budget setting of AL. We set the distance function for GSS across all experiments as $d_\gamma(\mathbf{x}_i, \mathbf{x}_j) = \max(0, d_{ij} - \gamma)$, where $d_{ij} = ||\mathbf{x}_i - \mathbf{x}_j||_F^2$. The distance function focuses on samples within the $\gamma$ radius, similar to ProbCover Yehuda et al. (2022), but with adding smoothness to values outside the radius $\gamma$.

We adopt two strategies in the AL rounds: 1) start straightforward from an empty label set with all rounds done by the AL method; 2) in the first round, labelled instances are selected using k-medoids with FasterPAM Schubert & Rousseeuw (2021); in other words, each AL method starts to work in the second round. For the larger datasets with the second strategy, we increase the budget in each round to 100 to reveal a clearer trend when labels increase. Note that this could still be considered low-budget compared to other methods where 10% of datasets is labelled for tuning parameters Yehuda et al. (2022). Because Silhouette is only valid when there are at least two labelled points, in the first strategy, we use k-medoids to pick the first 2 points for our method. All experiments were run with Dell XE9640 cluster system running Linux with Dual Xeon 36-core 8452Y compute nodes, 512GB of RAM and an Nvidia H100 GPU. We report the run time of all methods in the Appendix.

### 4.1 BASELINE METHODS

We compare our method to model-agnostic AL methods including: *ProbCover* Yehuda et al. (2022), *MaxHerding* Bae et al. (2024), *KernelHerding* Chen et al. (2012), *MaxHerding-nongreedy* Bae et al. (2024), and *Random*. We adopt the 1-NN classifier and focus on a low-budget AL setting without a validation set for hyper-parameter tuning. We follow the original papers to set hyper-parameters for each baseline. Previous experiments Yehuda et al. (2022); Hacohen et al. (2022); Bae et al. (2025) show that the model-agnostic AL performs better than interactive AL in low-budget settings and the latter requires a validation set to tune the model, so we exclude interactive AL here. We run 3 repetitions and plot the mean accuracy for *Random*. The rest of the methods are deterministic, so we report the Accuracy (ACC) from a single run. The baselines are (with $k(\cdot, \cdot)$ the RBF kernel):

- *Random* Randomly sampling at each iteration.
- *ProbCover* Pick the sample with the most number of $\gamma$-near neighbourhoods, where $\gamma$ is a pre-defined radius. Once a sample is selected, itself and its $\gamma$-near neighbourhoods are not considered in the next selection.
- *MaxHerding* Obtain the $(L + 1)$-th sample following $\tilde{\mathbf{x}}_{L+1} \in \arg\max_{\tilde{\mathbf{x}} \in \mathcal{U}} \frac{1}{N} \sum_{n=1}^{N} \max\{k(\mathbf{x}_n, \tilde{\mathbf{x}}) - \max_{\mathbf{x}' \in \mathcal{L}} k(\mathbf{x}_n, \mathbf{x}'), 0\}$.
- *KernelHerding* The $(L + 1)$-th sample is defined by $\tilde{\mathbf{x}}_{L+1} \in \arg\max_{\tilde{\mathbf{x}} \in \mathcal{U}} \frac{1}{N} \sum_{n=1}^{N} k(\mathbf{x}_n, \tilde{\mathbf{x}}) - \frac{1}{L+1} \sum_{l=1}^{L} k(\tilde{\mathbf{x}}_l, \tilde{\mathbf{x}})$.
- *MaxHerding-nongreedy* Choose a batch of $b$ samples by $\arg\max_{\mathcal{S} \subset \mathcal{U}, |\mathcal{S}| = b} \frac{1}{N} \sum_{n=1}^{N} \max_{x' \in \mathcal{L} \cup \mathcal{S}} k(\mathbf{x}_n, \mathbf{x}')$.

---

[2]The TrpB dataset is available at: https://github.com/csiro-funml/variationalsearch.

[3]The Phishing Websites dataset is available at the UCI Machine Learning Repository: https://archive.ics.uci.edu/.

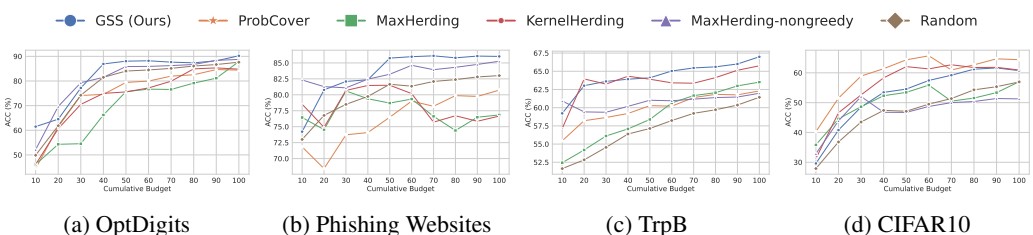

(a) OptDigits      (b) Phishing Websites      (c) TrpB      (d) CIFAR10

Figure 3: Accuracy (ACC) of GSS and baselines with all rounds done by AL.

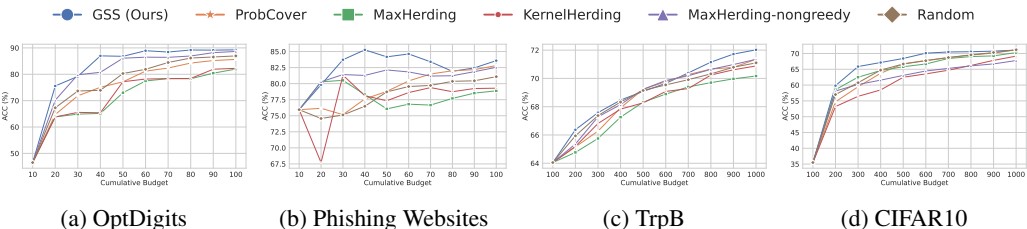

(a) OptDigits      (b) Phishing Websites      (c) TrpB      (d) CIFAR10

Figure 4: Accuracy (ACC) of GSS and baselines with the first round done by k-medoids.

## 4.2 RESULTS AND DISCUSSION

Overall, our proposed GSS method performs best in the experiments as presented in Fig. 3 and 4. In Fig. 3, with all rounds done by AL, GSS performs inferiorly compared to ProbCover and Kernel-Herding on CIFAR10. This is potentially due to the performance in the first round, where it performs second-worst. GSS involves the closest and second closest labelled points, and thus is more sensitive to noise in the initial round when labelled points are less informative. GSS consistently improves itself over increasing AL rounds. This demonstrates its robustness and generalisation capability, and supports our theoretical finding. In contrast, the performance of other methods exhibits notable variability, highlighting their lack of consistency across different data distributions. This underscores the sensitivity of these baselines to heuristic hyper-parameter choices and dataset-specific tuning. On the Phishing Websites dataset, all methods are struggling to keep improving as the number of labelled instances increases. This is potentially due to the small sample size, and categorical features therein make this even worse. The kernel-based method relies on the RBF kernel being effective. When dealing with categorical data, such as in Phishing Websites and TrpB, they exhibit a rather inconsistent performance, even decreasing as the number of labels increases.

Random method performs consistently across all datasets, because it samples unbiasedly from the underlying data distribution and the results are smoothed by averaging over 3 runs. The MaxHerding-nongreedy performs better than Random in the OptDigits dataset, while falling behind Random in CIFAR10. GSS is the only method that outperforms Random consistently. This aligns with previous findings Zhu et al. (2019); Siméoni et al. (2021), indicating that under low labelling budget conditions, achieving significant improvements over random selection remains challenging.

### 4.2.1 COMPARISON ON IMBALANCED DATASET

Imbalanced class distributions are prevalent in practical applications; we further assess the performance of GSS under such a situation by introducing controlled class imbalance into CIFAR10 dataset. We follow the previous research leveraging the long-tail imbalance generation algorithm Cui et al. (2019) to generate the imbalanced CIFAR10 dataset. All methods are initialised with 100 label samples selected by k-medoids. Figure 5 shows the results. We use the Matthews correlation coefficient (MCC) Chicco & Jurman (2020) as a performance indicator, considering that MCC has been proven as one of the best metrics to summarise performance under imbalanced data Powers (2020). All methods perform closely in the early few rounds, and GSS is leading the performance afterwards. The ability to consistently improve the performance between AL rounds relies on the ability to adjust hyper-parameters in-between.

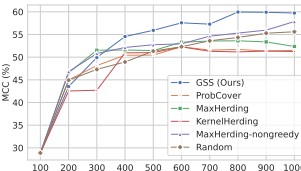

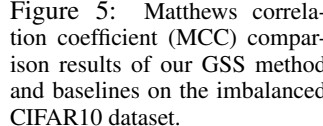

Figure 5: Matthews correlation coefficient (MCC) comparison results of our GSS method and baselines on the imbalanced CIFAR10 dataset.

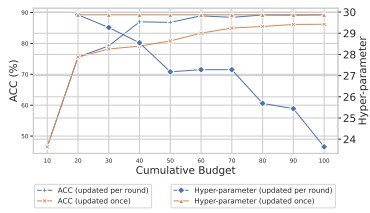

(a) Hyper-parameter updated per round vs. once

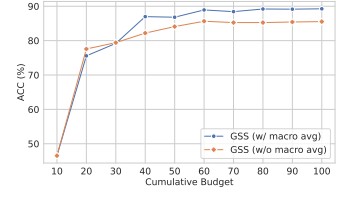

(b) W/ vs. w/o macro-average strategy

Figure 6: Performance of GSS on the OptDigits dataset under different settings.

### 4.3 ABLATION STUDY

To further validate the effectiveness of GSS, we conduct an ablation study on the OptDigits dataset. In the first round, samples are selected by k-medoids.

#### 4.3.1 HYPER-PARAMETER UPDATED EACH ROUND VS. ONCE

To show the advantage of dynamically updating the hyper-parameter we compare two variants of GSS: 1) find the optimal hyper-parameter in the second round, and then fix it in subsequent rounds (hyper-parameter updated once), and 2) find and update the optimal hyper-parameter in each round.

The results in Fig. 6(a) show that dynamically updating the hyper-parameter brings performance improvement over a static strategy. With hyper-parameter updated in each round, GSS is able to sense the change of labelled instance distribution, pick better hyper-parameter, and boost performance. Fig. 6(a) also shows that the hyper-parameter generally keeps decreasing as the cumulative budget increases. Intuitively, when labelled instances become more and more "crowded", the distance function tends to become more sensitive to capture finer details by decreasing $\gamma$.

#### 4.3.2 GSS WITH VS. WITHOUT MACRO-AVERAGE STRATEGY

We examine the performance of GSS with either macro-average strategy or micro-average strategy (w/o macro avg). With micro-average strategy, the medoid Silhouette is simply averaged across all unlabelled instances, without considering the number of instances in each cluster (similar to letting $|\mathcal{C}_l| = 1$ in Eq. equation 3). As shown in Fig. 6(b), when only 20 instances are selected, the macro-average strategy is not better than the micro-average strategy, likely because the number of labelled points is too small to capture useful information over noise. The performance of macro-average strategy improves fast and surpasses micro-average strategy after the cumulative budget is larger than 20, leaving a performance gap in-between, which shows the effectiveness of it.

## 5 CONCLUSION, LIMITATIONS AND FUTURE WORK

We consider the problem of identifying data to label by using an incremental clustering approach, instead of the more traditional active learning approaches. We introduce Greedy Silhouette Search, making the costly Silhouette metric feasible for automating active labelling under a low budget. It guides both sample selection and hyper-parameter configuration, saving us from requiring a labelled validation set for hyper-parameter tuning. The proposed GSS is supported by a bound on classification error (Theorem 1) and empirically on four benchmark datasets. Our automatic hyper-parameter choice (without any labelled validation dataset) results in an expected reduction in the effect radius.

GSS has its limitations similar to the original Silhouette. It prefers data distributed in a convex-shape, and may not perform well if the data clusters have irregular shapes or are of varying sizes, though we alleviate the latter by adopting a macro-averaging strategy. Our future research direction is to learn a metric or kernel to better represent the underlying data distribution.

## 6 REPRODUCIBILITY STATEMENT

The paper, appendix material, and attached code fully disclose all the information needed to reproduce the main experimental results of the paper, such that the conclusions of the paper can be reproduced by an external party.

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

## A  APPENDIX

### A.1  THE BOUND ON THE ACCURACY OF A 1-NN

MaxHerding Bae et al. (2024) defines their bound on the accuracy of 1-NN classifier $\hat{f}$ as:

$$
\mathbb{E}[1_{f(\mathbf{x}) \neq \hat{f}(\mathbf{x})}] \leq (1 - \mathbb{E}_{\mathbf{x}}[\max_{\tilde{\mathbf{x}} \in \mathcal{L}} k(\mathbf{x}, \tilde{\mathbf{x}})]) \\
+ \mathbb{E}_{\mathbf{x}}[\max_{\tilde{\mathbf{x}}: f(\mathbf{x}) \neq f(\tilde{\mathbf{x}})} k(\mathbf{x}, \tilde{\mathbf{x}})].
\tag{6}
$$

$\mathbb{E}_{\mathbf{x}}[\max_{\tilde{\mathbf{x}} \in \mathcal{L}} k(\mathbf{x}, \tilde{\mathbf{x}})]$ in the first term is the proposed "generalised coverage" to be maximised. The second term $\mathbb{E}_{\mathbf{x}}[\max_{\tilde{\mathbf{x}}: f(\mathbf{x}) \neq f(\tilde{\mathbf{x}})} k(\mathbf{x}, \tilde{\mathbf{x}})] > 0$ is data dependent and does not change with the AL process. In other words, it does not go better than the constant $\mathbb{E}_{\mathbf{x}}[\max_{\tilde{\mathbf{x}}: f(\mathbf{x}) \neq f(\tilde{\mathbf{x}})} k(\mathbf{x}, \tilde{\mathbf{x}})]$ even with number of labelled points going to infinity.

Recall that in **Theorem 1** of our paper, the expected error of the 1-NN classifier $\hat{f}$ is bounded from above by:

$$
\mathbb{E}[1_{f(\mathbf{x}) \neq \hat{f}(\mathbf{x})}] \leq (1 - \Pr_{\mathbf{x}}(\frac{\min_{\tilde{\mathbf{x}} \in \mathcal{L}} d(\mathbf{x}, \tilde{\mathbf{x}})}{\min_{\tilde{\mathbf{x}} \in \mathcal{L}}^{2} d(\mathbf{x}, \tilde{\mathbf{x}})} \leq \theta)) \\
+ \Pr_{\mathbf{x}}(\frac{\min_{\tilde{\mathbf{x}}: f(\mathbf{x}) \neq f(\tilde{\mathbf{x}})} d(\mathbf{x}, \tilde{\mathbf{x}})}{\min_{\tilde{\mathbf{x}} \in \mathcal{L}}^{2} d(\mathbf{x}, \tilde{\mathbf{x}})} \leq \theta).
\tag{7}
$$

Similarly, the first term in the RHS of the inequality is related to the medoid Silhouette to be optimised. Now consider $\Pr_{\mathbf{x}}(\frac{\min_{\tilde{\mathbf{x}}: f(\mathbf{x}) \neq f(\tilde{\mathbf{x}})} d(\mathbf{x}, \tilde{\mathbf{x}})}{\min_{\tilde{\mathbf{x}} \in \mathcal{L}}^{2} d(\mathbf{x}, \tilde{\mathbf{x}})} \leq \theta)$. The numerator is a constant similar to that in MaxHerding. With adding labelled samples, $\min_{\tilde{\mathbf{x}} \in \mathcal{L}}^{2} d(\mathbf{x}, \tilde{\mathbf{x}})$ is non-increasing, and can be decreasing if we sample $\mathbf{x}$ from a continuous distribution and not to add repeated labelled points. This gives rise to the desired asymptotic property when the number of labelled points increases.

### A.2  COMPARISON OF RUNNING TIME IN SECONDS

We report the running time of the experiments for the first strategy (start straightforward from an empty label set with all rounds done by AL methods). ProbCover is the most efficient non-random method because of the simple euclidean distance used therein, and the strategy to remove points gradually. MaxHerding is slower than KernelHerding because the introduced "max kernel function" involves extra max operation. The nongreedy version of it is even slower due to the complexity in set optimisation. GSS shows comparable complexity to KernelHerding and MaxHerding, and is faster as the number of AL round goes up.

Table 2: Running time on Phishing Websites dataset.

| Round | GSS (Ours) | Prob Cover | Max Herding | Kernel Herding | MaxHerding -nongreedy | Random |
|---|---|---|---|---|---|---|
| 1 | 3.5114 | 0.0240 | 4.0287 | 1.8295 | 58.5903 | 0.0012 |
| 2 | 3.6371 | 0.0227 | 3.8697 | 1.9387 | 56.7523 | 0.0013 |
| 3 | 3.0638 | 0.0215 | 3.7760 | 1.9336 | 85.6938 | 0.0013 |
| 4 | 2.8642 | 0.0221 | 3.8004 | 1.9306 | 58.0654 | 0.0013 |
| 5 | 2.7693 | 0.0216 | 3.6947 | 1.9293 | 57.7699 | 0.0013 |
| 6 | 2.8099 | 0.0214 | 3.6473 | 1.9329 | 59.5970 | 0.0013 |
| 7 | 2.8547 | 0.0211 | 3.6123 | 1.9502 | 59.8355 | 0.0012 |
| 8 | 2.7197 | 0.0212 | 3.5978 | 1.9370 | 89.9733 | 0.0013 |
| 9 | 2.3847 | 0.0213 | 3.5448 | 1.9186 | 89.6281 | 0.0013 |
| 10 | 2.9946 | 0.0221 | 3.5668 | 1.9124 | 60.0225 | 0.0012 |

Table 1: Running time on OptDigits dataset.

| Round | GSS (Ours) | Prob Cover | Max Herding | Kernel Herding | MaxHerding -nongreedy | Random |
|---|---|---|---|---|---|---|
| 1 | 1.0974 | 0.0199 | 0.7763 | 0.3988 | 10.8670 | 0.0005 |
| 2 | 1.0447 | 0.0217 | 0.8247 | 0.4006 | 10.6583 | 0.0006 |
| 3 | 0.6832 | 0.0214 | 0.7890 | 0.4103 | 10.7127 | 0.0006 |
| 4 | 0.6673 | 0.0213 | 0.7525 | 0.3903 | 10.5323 | 0.0006 |
| 5 | 0.7533 | 0.0227 | 0.7202 | 0.4087 | 10.5832 | 0.0006 |
| 6 | 0.6786 | 0.0213 | 0.7281 | 0.4104 | 10.5034 | 0.0006 |
| 7 | 0.6805 | 0.0212 | 0.7123 | 0.4287 | 10.4795 | 0.0006 |
| 8 | 0.6442 | 0.0218 | 0.7045 | 0.4319 | 10.5425 | 0.0006 |
| 9 | 0.6831 | 0.0215 | 0.6973 | 0.4149 | 10.3649 | 0.0006 |
| 10 | 0.7710 | 0.0214 | 0.6913 | 0.4006 | 10.4116 | 0.0006 |

Table 3: Running time on TrpB dataset.

| Round | GSS (Ours) | Prob Cover | Max Herding | Kernel Herding | MaxHerding -nongreedy | Random |
|---|---|---|---|---|---|---|
| 1 | 13.9187 | 0.1775 | 13.7053 | 6.0163 | 26.2610 | 0.0023 |
| 2 | 11.0669 | 0.2287 | 13.1580 | 6.4557 | 28.7492 | 0.0026 |
| 3 | 9.8945 | 0.0950 | 12.6008 | 6.4612 | 27.5707 | 0.0025 |
| 4 | 9.2774 | 0.3687 | 12.2238 | 6.6284 | 27.9558 | 0.0025 |
| 5 | 8.5968 | 0.1561 | 12.0822 | 6.3777 | 27.4611 | 0.0026 |
| 6 | 8.1340 | 0.2381 | 12.0328 | 6.4165 | 27.8891 | 0.0026 |
| 7 | 8.0193 | 0.2369 | 12.0449 | 6.5375 | 26.5342 | 0.0026 |
| 8 | 7.9281 | 0.1094 | 12.0381 | 6.4834 | 28.8684 | 0.0026 |
| 9 | 7.7948 | 0.1115 | 11.8515 | 6.4559 | 26.3114 | 0.0025 |
| 10 | 7.9075 | 0.0938 | 12.0690 | 6.4883 | 27.8513 | 0.0026 |

Table 4: Running time on CIFAR10 dataset.

| Round | GSS (Ours) | Prob Cover | Max Herding | Kernel Herding | MaxHerding -nongreedy | Random |
|---|---|---|---|---|---|---|
| 1 | 147.9980 | 0.6567 | 92.7664 | 47.5405 | 412.0120 | 0.0065 |
| 2 | 115.1394 | 0.6106 | 94.0380 | 50.9838 | 405.3778 | 0.0067 |
| 3 | 99.7956 | 0.5977 | 91.7539 | 51.0448 | 537.2294 | 0.0071 |
| 4 | 93.5841 | 0.5630 | 91.1137 | 51.6196 | 949.1273 | 0.0074 |
| 5 | 90.2527 | 0.5456 | 91.8859 | 51.5399 | 405.9316 | 0.0073 |
| 6 | 83.8906 | 0.5119 | 90.2955 | 51.4680 | 408.3155 | 0.0073 |
| 7 | 84.6417 | 0.5430 | 89.2639 | 51.7447 | 403.8474 | 0.0072 |
| 8 | 82.2778 | 0.4952 | 89.7910 | 51.8867 | 273.1389 | 0.0072 |
| 9 | 81.2419 | 0.4889 | 90.2507 | 51.9105 | 404.5455 | 0.0072 |
| 10 | 77.0552 | 0.4796 | 92.3652 | 51.7752 | 269.7230 | 0.0073 |

