# OpenReview forum: "Automating Active Labelling with Greedy Silhouette Search"
_ICLR.cc/2026/Conference — ICLR 2026 Conference Withdrawn Submission_

### Official Review · Reviewer_1HmQ · 2025-10-18

**Soundness:** 2
**Presentation:** 3
**Contribution:** 2
**Rating:** 4
**Confidence:** 4

**Summary:**

This paper introduces a greedy silhouette based strategy for automating active labeling, offering a clear and well-executed framework with solid experiments but limited theoretical depth and motivation.

**Strengths:**

1 The paper presents a clear framework that integrates clustering based reasoning with active labeling, and the overall method is easy to follow and reproducible.
2 The experimental evaluation is comprehensive, covering multiple datasets and comparing against a range of baselines, with results showing steady and interpretable improvements.

**Weaknesses:**

1. The paper provides very limited conceptual grounding for the problem it aims to solve. The introduction offers little discussion of what makes active labeling inherently challenging, why existing clustering‐based or uncertainty‐based methods fall short, or what specific gaps the proposed approach fills. The motivation is largely circular like this: since the silhouette metric has not been used for active learning before, the authors propose to make it usable through a greedy procedure. This rationale feels more like a technical curiosity than a problem driven innovation. A stronger motivation would require a clearer articulation of the real challenges in automating sample selection (e.g., representativeness vs. diversity trade-offs, stability of clustering objectives, computational bottlenecks) and a principled argument for why silhouette-based optimization specifically addresses them.
2.Theorem 1 decompose the 1-NN classification error into the ratio of nearest and second-nearest distances. To me, this bound holds without assumptions on the shape of p(y|x). It remains a descriptive identity rather than a generalization result. However, the proposed algorithm optimizes the empirical medoid-silhouette average and its greedy variant. The paper never proves that improving empirical silhouette correlates with increasing p(a/b – theta) or decreasing the second term in the population bound. Consequently, Theorem 1 and the algorithm live at different statistical levels. To close this gap, the paper would need some generalization or stability argument, for example, a uniform convergence or concentration bound linking empirical silhouette to its population counterpart, or a probabilistic guarantee that greedy silhouette maximization reduces the true risk.
3.The labeled set is built adaptively by the algorithm itself, and the hyperparameter gamma is tuned on the same empirical objective each round. This, as all AL method do,  breaks the i.i.d. assumption required for most generalization arguments: the sampling is on-policy and biased toward high-density or boundary regions, and gamma is over-fit to the same data. The paper provides no mechanism such as hold-out splitting, cross-fitting, or adaptive uniform bounds, to control this bias. A more rigorous analysis could rely on adaptive-data-dependent generalization theory or stability guarantees that bound how much the empirical improvement transfers to unseen data.
4.Because Theorem 1 imposes no margin, noise, or separability assumptions, the bound is necessarily loose and non-quantitative. The claim that the second term is non-increasing as labeled data grow is at best an intuition rather than guarantee to me. To obtain meaningful guarantees, one would need standard geometric or noise assumptions.
5.The theoretical analysis and the entire algorithm rely on 1-NN distance ratios and silhouette scores computed in a feature space. However, the paper provides no justification for why nearest-neighbor geometry remains meaningful in high-dimensional embeddings such as CIFAR-10. In such spaces, distances typically concentrate and lose discriminative power, which undermines both the theoretical bound (Theorem 1) and the empirical surrogate that depend on well-separated neighborhoods. The authors mention using pretrained ResNet features but offer no metric learning, normalization, or analysis to ensure the embedding preserves neighborhood structure. As a result, it is unclear whether the reported improvements stem from genuine geometric gains or from incidental properties of the chosen feature extractor.

**Questions:**

Could you clarify under what conditions the nearest neighbor distances and silhouette ratios remain meaningful in high dimensional feature spaces such as CIFAR-10, and how their framework mitigates the well-known problem of curse of dimensionality in such settings?

---

### Official Review · Reviewer_Uyyh · 2025-10-31

**Soundness:** 2
**Presentation:** 3
**Contribution:** 1
**Rating:** 2
**Confidence:** 5

**Summary:**

The paper addresses active learning in the very low-budget regime, a problem that has attracted growing attention in recent years. It adopts a clustering paradigm and proposes a greedy approach based on the silhouette coefficient - an established clustering criterion typically used to evaluate clustering quality. The method is compared with two recent coverage-based approaches shown to be effective in low-budget settings, ProbCover and MaxHerding; as in those works, a bound on the expected 1-NN error is provided. Empirical results on four datasets are mixed, as the method is not competitive on CIFAR-10 - the only shared dataset for which the comparison methods have also reported results in the original paper.

**Strengths:**

The paper proposes to use the silhouette coefficient for clustering-based query selection, a criterion not previously used for this purpose. Its empirical evaluation broadens the benchmark suite by including three datasets that are not commonly used in active learning studies.

**Weaknesses:**

The use of a clustering criterion for query selection is very similar to TypiClust (Hacohen et al. 2022, cited in this work), yet no direct comparison is provided. Instead, the empirical evaluation is limited to two coverage-based baseline methods that are more principled, using submodular objectives to justify greedy selection. All three approaches have a single distance-scaling hyperparameter -- ProbCover’s radius $\delta$, MaxHerding’s RBF width $\sigma$, and this paper’s threshold $\gamma$ (line 338). None require labeled validation data: this paper uses a golden-section search, ProbCover offers a self-supervised heuristic for $\delta$, and MaxHerding reports robustness to $\sigma$. It is therefore puzzling that the paper repeatedly claims not needing a labeled validation set as a contribution (e.g., line 149), while also asserting ``delicate and complex hyper-parameter tuning'' in the abstract -- neither of which characterizes the baselines they compare against.

While using the Silhouette coefficient is novel, the paper does not justify it convincingly. TypiClust already uses greedy k-means–based selection, which is preferable for speed and scale; this matters at modern unlabeled-pool sizes (millions of images), and even for CIFAR-10 with its 50k training images. A Silhouette-greedy approach could help when the Euclidean distance is not sufficient, but here the only distance flexibility is the threshold parameter $\gamma$. At minimum, the paper should compare this criterion against its TypiClust predecessor, and also compare its computational complexity to that of k-means.

**Questions:**

While evaluating a 1-NN classifier is justified by the theoretical guarantees, you should also include comparisons using standard embedding-space classifiers (e.g., SVMs).

---

### Official Review · Reviewer_JWRb · 2025-11-01

**Soundness:** 3
**Presentation:** 1
**Contribution:** 2
**Rating:** 4
**Confidence:** 3

**Summary:**

This paper introduces Greedy Silhouette Search (GSS), an active learning method for low-budget settings. GSS uniquely uses the Silhouette clustering metric as a single, self-contained criterion to simultaneously guide sample selection and automate hyper-parameter tuning. This approach eliminates the need for a labeled validation set, which is often unavailable in low-budget scenarios. Experiments show GSS achieves competitive performance compared to baselines that require extensive tuning.

**Strengths:**

1. The problem studied in this paper is valuable.
2. The proposed approach is effective.

**Weaknesses:**

1. This paper is really poorly written because the overall narrative isn't very clear, and there also seem to be some minor formatting errors.
2. The novelty of this paper is not very strong.
3. The assumption underlying Algorithm 3 is unsubstantiated.
4. The experimental validation is clearly insufficient.
    - The ablation study was only tested on a single dataset
    - There are no experiments demonstrating the efficiency improvements of the proposed algorithm.
    - Moreover, the experimental results do not show a significant improvement, and in some cases, it is not even the best.

**Questions:**

Given that golden-section search (Algorithm 3) relies on the unimodality assumption of the objective function for convergence , what theoretical or empirical evidence supports that the complex, discrete, and greedily-computed medoid Silhouette score (Sγ∗​) is unimodal with respect to γ, and how does the algorithm guarantee a robust solution if this assumption does not hold?

---

### Official Review · Reviewer_cped · 2025-11-01

**Soundness:** 2
**Presentation:** 2
**Contribution:** 2
**Rating:** 4
**Confidence:** 4

**Summary:**

The paper proposes an active learning strategy based on the Silhouette clustering metric. The motivation is to improve the practicality of clustering-based AL strategy by using a Greedy Silhouette Search. The paper also proves a generalization error bound based on 1-nearest neighbor classifier using the labeled set from GSS.

**Strengths:**

The paper identifies key issues with clustering-based AL methods, including complexity and sensitivity to hyperparameters. The greedy search is a reasonable solution to the problem. The algorithmic part is presented clearly. There is a specific strategy for optimization with hyperparameter, which aligns with the motivation. There are evaluation results on both balanced datasets and imbalanced datasets.

**Weaknesses:**

1. The scope of the paper is limited. Even though the paper introduces existing literature, most methods are not discussed or compared later on, making most part of the related work section (2.2.2 and 2.2.3) useless. The experiments only include comparisons with very basic AL strategies or clustering-based strategies. At a complexity similar to $U^2$, there must be other algorithms that consider the informativeness and representativeness at the same time.

2. Even just within the presented comparisons, the results are highly unstable and the proposed method suffers on some datasets.

3. Given the greedy nature of the algorithm, and the error bound in Theorem 1 relies on the current labeled set distances, there should be a better analysis on the result in order to understand the practical meaning of the error bound. Compared to other frameworks, such as Bayesian active learning or other uncertainty-based AL, where the bound is more relevant to the label complexity, the results here are not so interpretable.

**Questions:**

Please see Weaknesses.

---

### Note · Authors · 2025-12-03

I have read and agree with the venue's withdrawal policy on behalf of myself and my co-authors.